# Exploring Managers’ Insights on Integrating Mental Health into Tuberculosis and HIV Care in the Free State Province, South Africa

**DOI:** 10.3390/ijerph21111528

**Published:** 2024-11-18

**Authors:** Christo Heunis, Gladys Kigozi-Male

**Affiliations:** Centre for Health Systems Research & Development, University of the Free State, P.O. Box 339, Bloemfontein 9301, South Africa; kigozign@ufs.ac.za

**Keywords:** managers’ insights, mental health integration, TB, HIV, building blocks, barriers, facilitators

## Abstract

The integration of mental health (MH) services into tuberculosis (TB) and HIV care remains a significant challenge in South Africa’s Free State province. This study seeks to understand the perspectives of public health programme managers on the barriers to such integration and to identify potential strategies to overcome these challenges. Data were collected between February and October 2021 using qualitative methods including four individual semi-structured interviews and two focus group discussions with a total of 15 managers responsible for the MH, primary healthcare, TB, and HIV programmes. Thematic data analysis was guided by an adapted version of the World Health Organization’s “building blocks” framework encompassing “service delivery”, “workforce”, “health information”, “essential medicines”, “financing”, and “leadership/governance”. Additionally, the analysis underscored the crucial role of “people”, acknowledging their significant contributions as both caregivers and recipients of care. Managers highlighted significant concerns regarding the insufficient integration of MH services, identifying structural barriers such as inadequate MH management structures and staff training, as well as social barriers, notably stigma and a lack of family treatment adherence support. Conversely, they recognised strong management structures, integrated screening, and social interventions, including family involvement, as key facilitators of successful MH integration. The findings emphasise the need for a whole-system approach that addresses all building blocks while prioritising the role of “people” in overcoming challenges with integrating MH services into TB and HIV care.

## 1. Introduction

A prevalent issue confronting healthcare systems in numerous low- and middle-income countries (LMICs) is the significant disparity between the demand for mental health (MH) services and the availability of such treatments [1,2]. A primary objective of the World Health Organization’s (WHO) Comprehensive Mental Health Action Plan 2013–2030 is to provide integrated and responsive MH care services in community-based settings [3]. Integrated primary healthcare (PHC) is a complex and multifaceted concept [4]. The World Health Report 2005 [5] (p. 108) stated that the goal of integration is “to tackle the need for complementarity of different independent services and administrative structures, so as to achieve common goals”. The report outlined integration at three levels: at the patient level as “case management”; at the service delivery point where “multiple interventions are provided through one delivery channel”; and at the systems level by “bringing together the management and support functions of different sub-programmes, and ensuring complementarity between different levels of care” [5] (p. 109). Thus, integration influences programme financing, planning, and delivery, ultimately impacting the achievement of health goals. Integrated care aims to elevate the quality of care, resulting in better health outcomes and greater patient satisfaction throughout their treatment journey, with the added potential for cost savings. However, despite these worthy aspirations, putting integrated care into practice is fraught with complex challenges.

Health managers often face significant obstacles, such as poor information sharing, a lack of time and personnel to spur programmes, and confusion over who is responsible for patient care [6,7,8]. These issues make it difficult to seamlessly integrate care across various programmes, levels of care, and organisations. While achieving functional integration of health services is considered an attainable goal [9], it is contingent upon several critical intersecting capabilities within the health system. These include ensuring fully operational frontline health services, adequately trained and motivated healthcare personnel, access to appropriate technology, and the decentralisation of authority and decision-making processes to lower cadres of frontline managers and staff [9]. Decentralisation facilitates the adaptation of integration processes to suit local contexts.

The transition to democracy in post-apartheid South Africa requires more attention to MH [10]. However, the field of MH care continues to generate significant public health concerns [11]. The Mental Health Care Act No. 17 of 2002 [12] was founded on the basic tenets set out by the WHO in 1996 to guide MH care law [13]. By setting a deinstitutionalisation agenda on par with countries like Brazil [14], the Act requires users to be treated as close to their place of domicile and in the least restrictive manner possible. By implication, a “human rights-driven ethos in patient care” era had arrived in South Africa [15] (p. 69). The integration of MH care services into PHC services requires fundamental change in the health system, from “one of human rights violations and poor health outcomes associated with care delivered through psychiatric institutions” to “one which respects human rights and promotes good health outcomes and recovery through the delivery of [MH] care in the [PHC] system” (p. 5) [16]. The 2022 World Mental Health Report stated that “[MH] is a lot more than the absence of illness: it is an intrinsic part of our individual and collective health and well-being… Ultimately, there is no health without [MH]” [17] (p. vi).

South Africa has a growing burden of non-communicable diseases (NCDs) where deaths due to major NCDs such as cardiovascular diseases, cancer, diabetes, and chronic lower respiratory diseases increased by 58.7% over 20 years, from 103,428 in 1997 to 164,205 in 2018 [18]. Kenge (2022) thus questioned whether the hospital-based model of prevention and control of MH disorders is suitable [19] (p. 19). The prevention and control of NCDs—including MH disorders—might better be integrated into other existing health services, notably PHC services as the foundation of universal health care.

The National Mental Health Policy Framework and Strategic Plan (MHPF&SP) 2013–2020 [20] was implemented by the National Department of Health to generate momentum towards the transformation and integration of MH care. MH services are meant to be offered as a key component of the PHC Package. The MHPF&SP 2013–2020 guided provinces in promoting MH, preventing mental illness, and providing treatment and rehabilitation for diverse MH disorders. It directed that MH care services should be decentralised and integrated into existing programmes through task sharing at the PHC level, with a view to increasing prevention, screening, self-management, care, treatment, and rehabilitation. Specialist district MH care and community-based teams had to be established to provide psychological services with appropriate accreditation and staffing in line with the PHC Re-engineering Strategy [21]. Selected key staff in every PHC facility had to be trained to provide basic MH care services under routine supervision and mentoring, with referral as appropriate.

However, the recently introduced National Mental Health Policy Framework and Strategic Plan (MHPF&SP) 2023–2030 [22] recognises significant under-investment in MH care across South Africa. This has led to substantial disparities in the accessibility of services and resources for MH care, perpetuating the historical legacy of colonial and apartheid-era MH systems that heavily relied on psychiatric hospitals. While progress has been made with the integration of MH into general healthcare in some provinces, nationally, major shortfalls in human resources characterise MH care at the PHC level [23]. While most provinces endorse the importance of integrating MH into PHC and some PHC nurse training initiatives have been undertaken, there is urgent need to strengthen the MH training of general health staff.

It is also a concern that there are presently only five indicators for MH included in the District Health Information System, viz. “[MH] caseload”, “PHC mental disorders treatment rate-new”, “[MH] separation rate”, “[MH] involuntary admission rate”, and the recently added “child and adolescent attempted suicide rate”. A qualitative study investigating the use of newly developed, contextually appropriate MH indicators in PHC facilities revealed that their implementation was generally perceived as feasible across three South Asian and three Sub-Saharan African countries, including South Africa (e.g., service utilisation by disorder [psychosis, bipolar disorder, depression, alcohol use disorder, epilepsy, suicide attempts], follow-up, and referral) [24]. This feasibility was attributed largely to the simplicity of the forms and the ongoing support provided during the design and implementation stages.

Further regarding the new MHPF&SP 2023–2030, while the deinstitutionalisation of MH care has made some progress in South Africa, the process has been devoid of concomitant development of PHC and community-based MH care services. This has led to a high number of people living with MH disorders facing housing insecurity, being in prison, and being constantly faced by “revolving door patterns of care” [25] (p. 403). MH services for children and adolescents are severely lacking. Substance use disorders treatment services are also in short supply with weak coordination between sectors.

In 2016, an inquest into the deaths of 144 MH care users and the disappearance of 44 others, who had been treated at Life Esidimeni hospitals in Gauteng province, revealed that approximately 1500 MH care users were haphazardly discharged or transferred to inadequate facilities without proper documentation [26]. This led to the abrupt discontinuation of their treatment. This sad event underscored the severe lack of continuity in MH care in South Africa, with many individuals effectively “lost” within the system. Coupled with the absence of a national clinical programme guideline for MH, this represents a significant challenge to the country’s healthcare system [27].

A growing body of evidence highlights the challenges in MH policy implementation in South Africa [11,28,29]. MH is often perceived as being less prioritised, inadequately resourced, and insufficiently supported compared to other critical public health programmes, such as HIV and maternal and child health [29]. This deficiency in policy implementation exacerbates existing inequalities and inequities in public health. The overarching goal of deinstitutionalisation cannot be humanely achieved without the development and implementation of effective models for delivering MH services at the PHC level [10,30].

Although research in this area is expanding within developing countries, significant gaps remain in the literature, particularly concerning LMICs that face a combined affliction from both TB and NCDs, threatening population health and further straining already stressed health systems [31]. Individuals with poor MH often exhibit higher rates of TB and HIV compared to those without MH issues. Conversely, individuals with TB or HIV are more prone to MH problems [32], which can result in decreased treatment adherence and increased mortality rates [33]. This bidirectional relationship, particularly with depression, has been extensively documented in both TB and HIV contexts. Given the significant overlap in vulnerable populations, any strategy aimed at eradicating TB and HIV must comprehensively address the MH and psychosocial needs of affected individuals and those at risk. It is therefore clear that effective management of mental illness, TB, and HIV requires an integrated and coordinated approach, including mental and behavioural healthcare services.

Studies have highlighted the significance of understanding the factors that impact healthcare implementation from the perspective of managers [34,35]. In South Africa, the delineation of management roles, with provincial-level managers overseeing and district managers ensuring frontline service delivery, underscores the crucial role of “leadership/governance” in enhancing the effectiveness of the district-based PHC system and achieving shared objectives. The current study examines public health programme managers’ insights into the challenges of integrating MH services into TB and HIV care in South Africa’s Free State province. The research questions specifically sought to identify the barriers and facilitators perceived by these managers that influence the integration of MH into TB and HIV care, and to propose solutions to overcome these challenges. As far as could be established, this is the first qualitative study focusing on programme managers’ views on the integration of MH into PHC services in South Africa.

In line with previous research related to public health systems being strengthened in the Free State [36,37], this analysis was guided by the WHO’s health systems strengthening framework. This framework emphasises six key “building blocks” crucial to a functioning health system: “service delivery”, “health information”, “workforce”, “essential medicines”, “financing”, and “leadership/governance” [38,39]. As all the building blocks are regarded as fundamental, the framework offers no weighting of the importance of the individual building blocks.

However, uncertainty persists regarding the optimal use of the building blocks framework to address fragmentation and inefficiencies in public health systems in LMICs. Mounier-Jack et al. [40] recommend that researchers adapt the framework to suit specific contexts. Collectively, the building blocks constitute the structural elements of the health system. However, de Savigny and Adam [41] (p. 32) suggest that the “missing ingredient” in the building blocks framework may be the role of “people”, not only as mediators and beneficiaries but also as active agents driving the system. A qualitative study applying the building blocks framework to assess health systems strengthening in Zambia highlighted the central role of people across all building blocks [42]. While the WHO building blocks framework primarily addresses structural and systemic aspects of health systems, it does acknowledge and integrate the critical role of “people”, both as healthcare providers and as recipients of health services. Highlighting the role of people facilitates the adaptation of health services to better meet their specific needs, thereby enhancing their societal relevance. This approach aligns with the foundational principles of PHC, such as equity, social justice, participatory decision-making, and intersectoral collaboration.

In South Africa, conditions such as HIV [43], TB [44], as well as MH disorders [45] are highly stigmatised, profoundly influencing the daily experiences of both healthcare providers and clinic service users. To address the people-related barriers—such as stigmatisation and inadequate family support for treatment adherence—frequently highlighted by managers in the current study and to incorporate the corresponding facilitators, a decision was made to add a seventh component, “people”, to the building block analytical framework.

## 2. Materials and Methods

### 2.1. Design

The study utilised an exploratory qualitative design, including individual semi-structured interviews (SSIs) and focus group discussions (FGDs) with health programme managers operating at both the provincial and district levels within the Free State province’s public healthcare system. As stated, the study design was guided by the WHO’s health systems strengthening framework which highlights six “building blocks” of the health system: “service delivery”, “health information”, “workforce”, “essential medicines”, “financing”, and “leadership/governance” [38,39].

### 2.2. Setting

With a population of 2,964,412 in 2022 [46], the Free State is one of South Africa’s nine provinces. The province accounted for 4,038,494 PHC headcounts in the 2021/22 period, representing 4.8% of the national PHC headcount of 84,511,186 [47] (pp. 196–199). However, according to the Health Systems Trust, the province reported lower PHC utilisation rates compared to national averages, with 1.8 visits per year in 2019/20 versus 2.0 visits nationally, and 1.6 visits per year in 2021/22 versus 1.7 visits nationally. Furthermore, the PHC professional nurse clinical workload in the Free State was higher in 2021/22, with 23.4 clients per nurse per day, exceeding the national rate of 21.9.

Regarding TB, the Free State reported 5569 cases of drug-susceptible TB out of a total of 121,883 cases in South Africa in 2021 [47] (pp. 144–146), accounting for 4.6% of the national total. The province’s TB programme’s performance fell below national averages across several key indicators: TB treatment initiation rate for clients aged five years and older in 2021/22 (90.6% vs. 93.4% nationally), drug-susceptible TB client loss to follow-up rate in 2021 (14.3% vs. 13.0% nationally), drug-susceptible TB death rate in 2021 (13.4% vs. 8.3% nationally), drug-susceptible TB treatment success rate in 2021 (71.3% vs. 77.9% nationally), multi-drug resistant TB client death rate in 2020 (26.6% vs. 17.5% nationally), multi-drug resistant TB treatment success rate in 2020 (56.1% vs. 60.8% nationally), TB symptom screening rate for clients aged five years and older in 2021/22 (89.3% vs. 95.8% nationally), and extremely drug-resistant TB client loss to follow-up rate in 2020 (25.0% vs. 17.8% nationally).

Concerning HIV, the Free State accounted for 415,029 out of 7,975,940 people living with HIV in South Africa in 2022, representing 5.2% of the national total [47] (pp. 153–161). According to the Health Systems Trust, the HIV prevalence rate in the overall population in 2022 was 14.7% in the Free State, which was higher than the national prevalence of 13.5%. Nevertheless, the Free State’s HIV programme demonstrates robust performance relative to the national programme across several key metrics. These include antiretroviral treatment coverage in 2020 (80.9% vs. 75.0% nationally), clients remaining on antiretroviral treatment in 2022 (75.7% vs. 67.7% nationally), HIV viral load suppression in 2020 (70.6% vs. 64.0% nationally), and antenatal clients initiated on antiretroviral treatment in 2021/22 (97.7% vs. 95.0% nationally). However, the HIV testing coverage rate in the Free State in 2022 was notably lower at 75.5% compared to the national rate of 82.9%, suggesting lower provider screening for HIV—including HIV screening among patients with MH disorders—and initiation of testing.

In the realm of MH care, the Free State has reported a strikingly higher MH separation rate of 20.1%, representing the proportion of clients admitted to hospitals for MH-related conditions, in contrast to the national rate of 3.8% [47] (p. 127). Previous studies in this province have exposed a fragmented MH network that remains largely hospital-centric, with authority and influence concentrated in a single specialist psychiatric hospital [48]. As previously discussed, the MH information system in the province is markedly underdeveloped. This shortcoming has led to the exclusion of MH indicators from health programmes at the PHC level, including those targeting TB and HIV. This omission, in itself, highlights a significant lack of prioritisation of MH care management and services within the province.

### 2.3. Participant Sampling, Recruitment

A non-probability purposive sampling approach was used to select 15 MH, PHC, TB, and HIV programme managers across the provincial and district levels of the health system. The sample sizes were chosen to achieve data saturation rather than the generalisability of findings. The Head of Department of the Free State Department of Health granted us permission to access the participants, who were recruited with assistance from their line managers. A study authorisation letter was presented to the line managers, who were then asked to distribute the letter and invite eligible managers to participate in the study. The line managers informed the managers about the voluntary nature of participation and requested to share their contact information with the research team. Subsequently, appointments for the SSIs and FGDs were scheduled with the participants to ensure minimal disruption to their work routines and duties.

### 2.4. Data Collection

Semi-structured guides featuring open-ended questions were crafted to elicit comprehensive and nuanced responses from participants, thereby enabling a thorough exploration of their perspectives. The questions included: “What, in your view, are the major barriers to integrating MH into TB and HIV care” and “What, in your view, are the major facilitators of integration of MH into TB and HIV care?”

Data were collected between February and October 2021. Experienced fieldworkers conducted the SSIs, while the first and second authors led the FGDs. All interviews and discussions, lasting about one and two hours, respectively, were audio-recorded with participants’ informed consent.

Initially, the plan was to conduct all data collection in person. However, the COVID-19 pandemic and the resulting protective measures, such as travel restrictions and social distancing [49], required a shift in approach. A hybrid methodology was adopted, combining both in person and virtual interactions. The pandemic also led to many health managers being reassigned to COVID-19 response roles, complicating fieldwork coordination. As a result, activities were staggered to comply with COVID-19 guidelines and accommodate participants’ availability. Virtual interviews and conferencing became a common practice for qualitative data collection during the pandemic [49,50]. While both the FGDs in the current study were conducted in person, upholding guidelines for social distancing and personal protection, two of the SSIs took place remotely. Throughout the process, strict ethical standards were upheld to ensure the protection of participants’ rights and privacy, regardless of the data collection method used [50].

### 2.5. Data Analysis

The discussions and interviews were audio-recorded and transcribed, and thematic analysis was performed by two independent coders following the methodology established by Braun and Clarke [51]. The study utilised the WHO health systems strengthening building blocks framework as its analytical lens [38,39,40,41,42], applying the individual blocks as predefined themes. Through careful coding, subthemes were identified and categorised into barriers and facilitators to integrating MH into TB and HIV care. To ensure the trustworthiness of the findings, data integrity principles were rigorously upheld, including triangulating data from both interviews and FGDs. Furthermore, a third coder reviewed the subthemes iteratively and engaged in discussions with the primary coders to achieve consensus.

### 2.6. Ethics Statement

This study received ethical clearance from the Health Sciences Research Ethics Committee at the University of the Free State (UFS-HSD2019/1574/2611-0003). Written informed consent was obtained from participants in data collection and handling aligned with the Declaration of Helsinki [52].

## 3. Results

This section presents the insights of provincial and district public health programme managers on the integration of MH into TB and HIV healthcare services within the Free State province. A total of 15 managers responsible for the MH, PHC, TB, and HIV programmes participated in the study: four in the individual SSIs and eleven in two FGDs, respectively, involving five TB programme and six HIV programme managers. Demographically, eight of the FGD participants were female and six were aged between 51 and 60 years. Five of the respondents had over 15 years of experience in their positions, with one respondent not specifying their experience. Additionally, seven participants held a bachelor’s degree. As depicted in Table 1, the identified subthemes underscored both the barriers impeding and the facilitators promoting the integration of these services across the WHO building blocks, with the inclusion of “people”, as perceived by the respondents.

### 3.1. Service Delivery

#### 3.1.1. Service Delivery Barriers

The first barrier perceived to hinder the integration of MH into TB and HIV services under the pre-defined “service delivery” theme was lack of MH support for diseases other than HIV, explained as:

*“For our HIV patients we do have the buddy system where we encourage them to have someone to talk to… But when we look at other diseases that are also deadly as HIV, we really don’t give that support”*.(HIV FGD participant 6)

The second perceived barrier under “service delivery” was the limited role of non-governmental organisations (NGOs) and community-based organisations (CBOs), articulated as:

*“[NGOs and CBOs] are crucial in terms of taking long term care and making sure that the patients adhere to treatment and go for follow up visits to [PHC] or to the specialist… and I know in our province we don’t have enough”*.(HIV FGD participant 4)

The third perceived barrier related to the “service delivery” building block was the communication gaps between initiating and receiving facilities during down referral. A participant observed:

*“When you discharge a [MH] patient from a psychiatric complex; they just discharge the patient, but they don’t report to the clinic to say, ‘today we discharged so and so. You must have a follow-up at your clinic on these dates.’ If that back referral is not done actively we will miss them. The problem is that they sometimes give the discharge letter to the patients in their hands, but they do not go back to the clinic… The down referral sometimes breaks the system and that causes the defaulters”*.(SSI participant 2)

In summary, the service delivery barriers observed by the managers included a lack of MH support for non-HIV diseases, insufficient involvement of NGOs and CBOs in patient care, and communication gaps during patient discharge that lead to missed follow-ups and treatment defaults.

#### 3.1.2. Service Delivery Facilitators

The first facilitator perceived to advance the integration MH into TB and HIV services related to the “service delivery” building block was to decentralise/deinstitutionalise MH care. A participant remarked:

*“The institutions that are decentralising [multi-drug resistant TB] patients need to be supported because as patients are decentralised from the institution they should also have some plan in terms of how are they going to be supported in terms of [MH]”*.(HIV FGD participant 6)

The second perceived facilitator related to “service delivery” was to establish a proper MH management structure. This was emphasised by a participant who stated:

*“Currently the [MH] directorate consists of two people. And it is really struggling in terms of resources. It used to have a complete structure from provincial level, district level and subdistrict and obviously it goes together with the budget so that we can have specific activities aiming to address specific areas of concern”*.(HIV FGD participant 4)

The third perceived facilitator under “service delivery” was to establish support groups for TB-HIV-MH patients. This was elaborated as follows by a participant:

*“I think we should not undermine the impact of a support group. In our adherence guidelines which is cutting across all, it is not only for HIV/TB, but also for [MH] and other chronics, there is what we call enhanced counselling. Where people need some extra support and there is also the support group issue where you can put people in a group to relate to each other and get some extra support. We found that over the years it is a really good idea”*.(HIV FGD participant 7)

To summarise, the managers believed that advancing the integration of MH into TB and HIV services required decentralising MH care, establishing a robust MH management structure, and creating support groups for patients.

### 3.2. Workforce

#### 3.2.1. Workforce Barriers

The first barrier perceived to hinder the integration of MH into TB and HIV care, related to the “workforce” building block, was deficits in staff MH training. This was articulated as follows by participants:

*“Most colleagues feel that [MH] is a specialised disease which is not applicable at their level, especially in the [PHC] setting. So, it is going to take some time to get people to understand that there are certain things in [MH] which can be treated at the level of [PHC]”*.(TB FGD participant 4)

*“I believe that the integration level is very minimal mainly due to lack of skills within our PHC setting. The majority of nurses are not clued up with [MH]. And we know there are limited clinicians who are specialists in that area of [MH]. Hence, the integration level is very slow”*.(HIV FGD participant 3)

The second perceived barrier related to the “workforce” building block was staff not prioritising MH, reflected on as:

*“You know, it is only a few who understand the priority; that [MH] should be prioritised. But I can say most of them don’t see it that way”*.(SSI participant 1)

The third perceived barrier under “workforce” was lack of MH specialist teams, shared as:

*“We are supposed to have more specialist teams but due to financial implications we could not appoint them yet”*.(SSI participant 2)

Briefly, the major workforce-related barrier was the lack of MH training among staff. Reportedly, many healthcare workers viewed MH as a specialised field not relevant to their roles in PHC, leading to a slow integration process. They needed time to understand that some MH issues can be managed at the PHC level. Another important barrier was the lack of prioritisation of MH. The managers observed that only a few healthcare workers recognised its importance, while most did not see it as a priority. Additionally, financial constraints were seen to have prevented the appointment of necessary MH specialist teams, further hindering integration efforts.

#### 3.2.2. Workforce Facilitators

The main facilitator perceived to advance the integration of MH into TB and HIV care related to the “workforce” building block was to improve staff MH training and retraining. Participants motivated this as follows:

*“They should be skilled [in] how to care for [MH] care users, how to manage the down referrals and the importance of our [MH] care users to adhere to treatment […] I think skilling of our professionals at [PHC] level will really be helpful”*.(SSI participant 1)

*“Lately I have realised that instead of doing more campaigns for communities, we need to re-train our health professionals because I think there is lack of skill regarding the provision of the service. That is our biggest need, retraining”*.(SSI participant 2)

The key facilitator for integrating MH into TB and HIV care perceived by the managers was to enhance staff training and retraining. Participants emphasised the need for healthcare professionals to be skilled in managing MH care users, handling referrals, and ensuring treatment adherence. They highlighted that retraining health professionals was more crucial than community campaigns due to a significant skill gap in service provision.

### 3.3. Health Information

#### 3.3.1. Health Information Barriers

Regarding the “health information” building block, the primary barrier perceived to impede the integration of MH into TB and HIV care was unawareness/non-use of the TB-HIV-MH screening tool. This was articulated by a participant as follows:

*“We should start to insist that most of the screening tools must be used. It could have been that they have only been concentrating on TB and HIV and neglecting the MH screening tool. So, it should be insisted and given more impetus to be able to get a better outcome”*.(TB FGD participant 4)

In summary, the main perceived health information-related barrier to integrating MH into TB and HIV care was the lack of awareness and use of the TB–HIV–MH screening tool. Participants emphasised the need to prioritise and consistently use this tool to improve outcomes.

#### 3.3.2. Health Information Facilitators

The corresponding principal facilitator identified by respondents for advancing the integration of MH into TB and HIV care was to integrate screening for TB, HIV and MH and to formalise the inclusion of MH data elements into TB-HIV-MH screening. This was elaborated by participants as follows:

*“I think integration of the screening tool at the entry level of each PHC facility should be introduced because some facilities are not aware of that H form that is integrated with all the diseases to screen”*.(TB FGD participant 5)

*“[CHWs] have already been trained on [MH]. They do it already. But you see… again, it was not a formal data element. You must have a formal data element to get it done consistently”*.(SSI participant 2)

The main health-information-related facilitator for integrating MH into TB and HIV care perceived by the managers was the need for consistent combined screening for TB, HIV, and MH at PHC facilities using the available tool that many healthcare workers do not seem to be aware of. Additionally, formalising the inclusion of MH data elements in these screenings was seen to be crucial for consistency. Participants highlighted the need for awareness and formal processes to ensure effective integration and consistent care.

### 3.4. Essential Medicines

#### 3.4.1. Essential Medicines Barriers

Regarding the “essential medicines” building block, while the managers did not express specific concerns about the availability of and access to medications for MH, TB, and HIV, they did highlight challenges related to treatment support. Specifically, they voiced concerns about the lack of nutritional support to patients on treatment. As one participant remarked:

*“As they are on treatment, they are complaining of hunger because they are not working. So, I think food parcels are the better solution for them”*.(TB FGD participant 1)

In summary, while managers did not report issues with the availability of essential medicines for MH, TB, and HIV, they did express concerns about the lack of nutritional support for patients undergoing treatment. They suggested providing food parcels to address patients’ complaints of hunger.

#### 3.4.2. Essential Medicines Facilitators

The principal facilitator identified for the “essential medicines” building block was to utilise CHWs to distribute medication to patients’ homes. As one participant advised:

*“If they did not come for their treatment the [CHWs] go to them to give them their treatment and advise them that they should take their treatment regularly. It is an ongoing service that they do in the community”*.(TB FGD participant 3)

To put it briefly, the main strategy perceived by the managers to ensure access to essential medicines was using CHWs to deliver medications directly to patients’ homes.

### 3.5. Financing

#### 3.5.1. Financing Barriers

The primary barrier perceived to hinder the integration of MH into TB and HIV services, related to the “financing” building block, was insufficient resources and a lack of dedicated funding for MH. This issue was conveyed as follows:


*“Normally when they talk about the budget, they prioritise HIV and TB”.*
(SSI participant 3)

The main financial barrier to integrating MH into TB and HIV services perceived by the managers was the lack of resources and dedicated funding for MH. This was often because budgets prioritise HIV and TB over MH.

#### 3.5.2. Financing Facilitators

The main facilitator perceived to advance the integration of MH into TB and HIV care under the “financing” building block was to utilise NGOs and CBOs more effectively. This was articulated as follows:

*“I think we must have more funding available for NGOs to render specific services. For example, we must license NGOs to accommodate [MH] care users… but we don’t find them to become compliant. There are thirteen criteria, norms and standards that they must meet before we can license them. [But] how can you enforce an NGO or a service provider to become compliant, but you don’t fund them?”*.(SSI participant 3)

As perceived by the managers, in order to better integrate MH into TB and HIV care, it is crucial to utilise NGOs and CBOs more effectively. This requires increasing funding for these organisations so they can meet compliance standards and provide necessary services. Without sufficient funding, NGOs and CBOs struggle to be compliant and support MH care users.

### 3.6. Leadership/Governance

#### 3.6.1. Leadership/Governance Barriers

The primary barrier perceived to hinder the integration of MH–PHC–TB–HIV services, related to the “leadership/governance” building block, was the limited MH management structure and lack of subdistrict MH managers. This was elucidated as follows:

*“The structure of the [MH] directorate at the provincial level is very slim and it is not approved as yet”*.(HIV FGD participant 1)

The main perceived obstacle to integrating MH into TB and HIV services was the weak leadership and governance structure. Specifically, the managers remarked on the limited MH management framework and a lack of MH managers at the subdistrict level. This issue was highlighted by the fact that the MH directorate at the provincial level was understaffed and not yet officially approved.

#### 3.6.2. Leadership/Governance Facilitators

The primary facilitator identified for advancing the integration of MH–PHC–TB–HIV services within the “leadership/governance” building block was to adopt a multisectoral approach to improve MH care provider training. This was conveyed as follows:

*“Capacitation should not be left on the shoulders of the [Department of Health]. This should be a multidisciplinary approach from all government, private, universities, etc.”*.(HIV FGD participant 6)

The key facilitator perceived by the managers for integrating MH–PHC–TB–HIV services under the “leadership/governance” framework was to adopt a multisectoral approach to enhance MH care provider training involving collaboration across government, private sectors, and universities, rather than relying solely on the Department of Health.

### 3.7. People

#### 3.7.1. People Barriers

The first barrier perceived to hinder the integration MH–PHC–TB–HIV services related to the added component, “people” was stigmatisation, explained as:

*“I think within our communities [MH patients] are perceived as outcasts. Even their immediate family members also perceive them as outcasts because they lack understanding of what this person is going through; what is happening to this person”*.(HIV FGD participant 3)

The second perceived barrier related to the added component was lack of family treatment adherence support, elaborated as:

*“[Sighs], it is a very difficult thing. Families and caretakers are dealing with [MH] patients every day. But I don’t think they give support to the user that they must get. A simple example, if they can ensure that [MH] care patients or users maintain their treatment they will not end up in defaulting. The moment they default treatment they end up in hospital systems”*.(SSI participant 2)

The third perceived barrier related to “people” was patients refraining from taking responsibility for their own health, clarified as:

*“What about your own commitment and your own responsibility to take your treatment? And that is not only for TB but for any patient. If you don’t take your treatment it is not the professional nurse’s, social worker’s responsibility. You must take your own responsibility…”*.(SSI participant 2)

In summary, stigmatisation was perceived to represent a significant barrier, as MH patients are often viewed as outcasts by their communities and even their families, who lack understanding of their conditions. Another perceived factor was the lack of family support for treatment adherence. Families and caretakers often struggle to help MH patients maintain their treatment, leading to treatment default and hospitalisation. Additionally, patients frequently fail to take responsibility for their own health. This issue is compounded by fears related to other conditions, such as TB, which can result in neglect of MH treatment.

#### 3.7.2. People Facilitators

The main facilitator perceived to advance the integration MH–PHC–TB–HIV services related to the added component, was to increase family involvement, voiced as:

*“We also encourage, for instance if it’s an elderly or a mentally challenged person, the family to be part of the medical consultations or whatever that will be discussed. Because if we are talking to a mentally challenged patient the message won’t be clear”*.(HIV FGD participant 5)

Another perceived facilitator related to the “people” building block was to partner with traditional and religious leaders, as depicted by a participant:

*“There is another element that we are forgetting. The traditional health people and the religious people. Your pastors and the inyangas in the community. Most of our people receive advices from religious or traditional help. And we need to capacitate them in terms of not only treatment adherence but also the mental support. We cannot run away from that because our communities do consult them”*.(HIV FGD participant 4)

To summarise, the primary “people”-related facilitator for integrating MH–PHC–TB–HIV services according to the managers was to increase family involvement in medical consultations, especially for elderly patients or patients with intellectual disabilities, to ensure clear communication. Additionally, partnering with traditional and religious leaders was seen as crucial for supporting treatment adherence and MH within the community.

## 4. Discussion

This study examines the barriers and facilitators influencing the integration of MH into TB and HIV care as perceived by the Free State public health programme managers within the framework of the health system building blocks. The argument ultimately posits that a whole-system approach—encompassing all six building block elements, with an added emphasis on the “people” component—is crucial for the successful integration of MH services into these healthcare domains.

### 4.1. Service Delivery

#### 4.1.1. Service Delivery Barriers

Lack of MH support for diseases other than HIV: A significant barrier to the integration of MH care, as perceived by managers in the current study, was the disparity in support provided to patients with MH conditions compared to those with HIV. While mechanisms like the “buddy system” for HIV patients were well-established, similar support structures for other serious conditions, including MH disorders, were conspicuously lacking. This imbalance undermined the provision of comprehensive care for patients with multimorbidities, underscoring the pressing need for a more equitable and integrated support system. The perceived barrier, “lack of MH support for diseases other than HIV”, identified by Free State managers is consistent with the directives of South Africa’s Mental Health Care Act of 2012 [12]. The Act designates PHC as the primary entry point for MH services and prioritises the integration of MH care into general health services. The MHPF&SP 2023–2030 [22] further underscores the urgent need to strengthen this integration.

Limited involvement of NGOs and CBOs: The primary and community-based MH care systems in South Africa are both underfunded and under-resourced [53]. The Free State managers pointed out that the limited involvement of NGOs and CBOs represented a major challenge in integrating MH into TB and HIV services. These organisations were perceived to be vital for providing long-term MH care and ensuring that patients adhere to their treatment plans. The perceived barrier was consistent with the principles outlined in the MHPF&SP 2013–2020 [20] (pp. 19–20), which stresses the necessity of intersectoral collaboration, including partnerships with NGOs, to address the social determinants of health.

Communication gaps during down referral: Effective communication between initiating and receiving healthcare facilities is essential for seamless patient management, particularly during the down-referral process. The National Referral Policy [54] (p. 16) stipulates that both the referring facility and the receiving facility must document all outgoing and incoming referrals in a designated referral register. The communication gap observed by the Free State programme managers, where discharges from psychiatric facilities reportedly often occurred without proper notification to follow-up clinics, likely significantly contributed to patient non-adherence and systemic failures. This issue highlighted the critical need for integrated protocols and systematic approaches to ensure that patients were consistently tracked and supported post discharge.

#### 4.1.2. Service Delivery Facilitators

Deinstitutionalising and decentralising MH care and supporting CBOs: The managers in the current study identified deinstitutionalising and decentralising MH care, along with enhancing the role of CBOs, as pivotal components in the broader strategy to integrate MH care into PHC. This perspective aligns with the MHPF&SP 2023–2030, which advocates for the gradual downsizing and replacement of long-term custodial specialist MH facilities with a comprehensive network of community-based residential and day care centres. However, the policy stipulates that deinstitutionalisation should only proceed once these essential community-based facilities are fully established, a requirement reportedly unmet in the Free State province. In 2009, Petersen et al. [55] noted that, akin to other LMICs, South Africa’s efforts to achieve deinstitutionalisation and comprehensive, integrated MH care was hindered by the lack of MH resources within the PHC Package and the inefficient utilisation of existing resources.

Establishing a proper MH management structure: The Free State managers’ recommendation to establish a robust MH management structure as a critical facilitator for integrated care aligns with the MHPF&SP 2013–2020 directive [20] (p. 31) to strengthen the capacity of district health management teams in planning, implementing, supervising, monitoring, and evaluating MH programmes at both district and community levels. Previous research in the Western Cape province has highlighted the pivotal role of district facility managers in developing and executing district MH information systems and policies, which are essential for improving health service outcomes at the district level [56].

Establish support groups for TB–HIV–MH patients: The Free State managers’ recognition of the importance of the role of social support, particularly through self-help groups, as a facilitator of MH improvement is strongly supported by research. For instance, a study in Myanmar found that participation in self-help groups served as a protective factor against depressive symptoms among people living with HIV [57]. Similarly, research in Zimbabwe on vulnerable young mothers revealed that engagement in self-help groups enhanced MH by strengthening peer support and fostering hope for the future [58]. In the United Kingdom, a study on men’s experiences with peer support groups for managing mental distress found that these groups provided a secure environment—a “safe space”—where men could challenge traditional masculine norms by exchanging personalised MH support and adopting specific roles within the group [59].

### 4.2. Workforce

#### 4.2.1. Workforce Barriers

Staff MH training deficits: The issue of inadequate MH training among PHC nurses, as highlighted by the Free State managers, aligns with findings of a study to assess MH literacy of PHC workers in South Africa and Zambia showing moderate MH literacy, which may hinder their ability to adequately recognise MH conditions, but with a wide range from low to high MH literacy [60]. The findings in the Free State also resonate with a recent international literature review that identified significant MH knowledge gaps and learning needs among PHC nurses [61]. This review underscored the necessity for PHC nurses to actively identify their MH learning needs and engage in targeted education to adequately prepare them for meeting the increasing demands for MH services.

Staff not prioritising MH: Not considering MH as one of the priorities within district health services has also been reported in a study on district managers’ perspectives of MH information processing and utilisation at the PHC level in the Western Cape [56]. This oversight reflects a broader issue within the health system where MH is often marginalised despite its critical impact on other public health challenges. Staff must be attuned to the fact that beyond stigma and social isolation, mental illness persists as a “silent driver of the global TB epidemic” [62].

Lack of MH specialist teams: As reported by the managers, the Free State province, which consists of five districts, had only one district MH specialist team. This is despite the MHPF&SP 2013–2020 [20] (p. 23) directive that district specialist MH teams should be established to support non-specialist PHC staff and community-based workers. The new MHPF&SP 2023–2030 [22] (p. 36) emphasises that district MH specialist teams are responsible for building capacity among users—patients and their families—to provide appropriate self-help and peer-led services, such as support groups facilitated by NGOs.

#### 4.2.2. Workforce Facilitators

Improving staff MH training and retraining: The approach suggested by the Free State managers aligns with the MHPF&SP 2023–2030 [22] (p. 28), which mandates that all health staff receive basic MH training, including anti-stigma education, along with regular supervision and mentoring. Provincial Departments of Health are tasked with expanding the MH workforce, and a task-shifting approach is emphasised, where trained non-specialist workers provide evidence-based psychosocial interventions under the guidance of specialists [22] (p. 28).

Re-establishing district MH specialist teams: This perceived facilitator of the Free State managers resonates with the MHPF&SP 2023–2030 [22] (p. 24) emphasis on the importance of district MH specialist teams. These teams are tasked with developing and implementing district MH plans, utilising all available resources within the district, and incorporating tools and lessons from South African innovations in district MH plans.

### 4.3. Health Information

#### 4.3.1. Health Information Barriers

Non-awareness/use of TB–HIV–MH screening tool: This perceived barrier corroborates the findings of a study on the views of facility managers on the use of MH information for planning services in the Western Cape province showing that MH information processing systems were fragmented and inadequate for decision-making [56]. According to this study, a lack of knowledge in information processing and utilisation, as well as poor information infrastructure and networking were associated with poor understanding about MH, not considering MH as one of the priorities within the district health services, and a lack of higher officials’ interest in the development of the MH programme.

Surprisingly, the Free State managers did not seem to be aware of two recognised screening guidelines: firstly, the 2023 Adult Primary Care guideline [63]—a prescribed guideline in South Africa in PHC settings for the screening, assessment, and management of diseases, and secondly, the Mental Health Gap Action Programme (MHGaP), recommended by the WHO for use by healthcare professionals [1].

#### 4.3.2. Health Information Facilitators

Integrating screening for TB, HIV, and MH: This proposed facilitator aligned with the MHPF&SP 2013–2020 [20] (p. 29) call for routine screening and treatment of physical illnesses during all consultations for individuals with mental illness. The new MHPF&SP 2023–2030 [22] (p. 24) again emphasises that MH interventions will be included in the core package of district health services. This should involve a task-sharing approach where trained non-specialist workers deliver evidence-based psychosocial interventions, including routine screening for mental illness during pregnancy and for other identified high-risk groups, along with a stepped approach to management and referral.

### 4.4. Essential Medicines

#### 4.4.1. Essential Medicines Barriers

Lack of nutritional support for patients on treatment: This perceived barrier resonates with a study in the USA showing that medically appropriate food support may improve MH for people living with HIV [64]. As shown in a review of studies in Africa that included co-infected adults and children, nutritional support is also important to minimise the harmful effects of food insecurity in HIV-TB populations [65]. This underscores the importance of integrating nutritional support into healthcare strategies to enhance MH outcomes and address the broader impacts of food insecurity on individuals with HIV–TB co-infection.

#### 4.4.2. Essential Medicines Facilitators

Utilising CHWs to take medicine to patients’ homes if they do not collect it from the clinic: This suggested facilitator is supported by a study in the Western Cape province that showed that home delivery of medication by CHWs was feasible at scale and affordable [66]. The integration of CHWs into home delivery of medication demonstrates significant potential to enhance medication adherence and access.

### 4.5. Financing

#### 4.5.1. Financing Barriers

Insufficient resources and a lack of dedicated funding for MH: According to the Free State managers, insufficient resources and inadequate dedicated funding for MH presented substantial challenges. These perceptions are consistent with the MHPF&SP 2013–2020 observation that inadequate MH financing significantly impedes progress [20] (p. 13). Research carried out across six sub-Saharan African countries, including South Africa, and South Asian nations, highlights critical obstacles to sustainable MH financing, including inadequate funding levels, pervasive inequalities in access, and entrenched poverty [67].

#### 4.5.2. Financing Facilitators

Utilising NGOs and CBOs effectively: As suggested by the Free State managers, effective use of NGOs and CBOs is crucial in the delivery of decentralised MH services. Despite government subsidies allocated to approximately 2000 MH NGOs in South Africa, the lack of coordination between government departments and these NGOs has resulted in fragmented care delivery [68].

### 4.6. Leadership/Governance

#### 4.6.1. Leadership/Governance Barriers

Limited MH management structure and lack of subdistrict MH managers: In South Africa’s district health management context, which is continuously shaped by major healthcare reforms from the National Department of Health, many managers lack the competencies required for effective leadership [69]. Research has shown that healthcare managers within districts are often selected from health professional backgrounds. While this provides valuable contextual insight, they frequently lack formal management training, thereby limiting their managerial effectiveness. A 2015 study on health system governance aimed at supporting integrated MH care in South Africa highlighted significant deficiencies in managerial and planning capacities at both provincial and district levels, particularly regarding the development and implementation of integrated MH care plans [70].

#### 4.6.2. Leadership/Governance Facilitators

Adopt a multisectoral approach to improve MH care provider training: Respondents in the current study emphasised the need for a multisectoral approach to training MH care providers at the PHC level. They argued that responsibility for this training should extend beyond the Department of Health to include universities and the private sector. Given the increasing demands of MH care, there is a critical need to fundamentally revise undergraduate nursing education to better prepare students for MH care delivery, particularly within the TB and HIV programmes. However, the South African Nursing Council has significantly reduced the content and clinical hours in undergraduate nursing curricula, focusing narrowly on foundational theory and basic interpersonal skills. As a result, newly qualified nurses are inadequately prepared to provide effective MH care.

### 4.7. People

#### 4.7.1. People Barriers

Stigmatisation: Respondents in the current study reported that MH patients are often viewed as “outcasts”, a perception consistent with findings by Mathias et al. (2024) [68], which reveal that in Ghana, India, and South Africa, mental illness is frequently interpreted through religious and traditional frameworks, attributing it to moral failings, taboo violations, or malevolent forces such as sorcery. These interpretations reinforce conservative moral codes and perpetuate stigma. The pervasive stigma surrounding mental illness often discourages individuals from seeking care due to fear of judgement and discrimination. This reluctance significantly contributes to the underutilisation of MH services, presenting major obstacles to the effective integration of these services into TB and HIV care settings.

Lack of family treatment adherence support: From the perspective of the Free State programme managers, the lack of family support for treatment adherence emerged as a significant barrier to integrating MH services with TB and HIV care. Inadequate family support can severely impact the management of MH disorders, TB, and HIV. Research consistently underscores the critical role of family involvement in improving treatment outcomes. For example, a study in China found that TB patients who received regular supervision and spiritual encouragement from family members demonstrated higher adherence to treatment protocols [71]. Similarly, research in Uganda showed that strong family cohesion was strongly associated with greater self-efficacy in adhering to antiretroviral treatment [72]. Furthermore, a multi-ethnic Asian cohort study observed that significant psychological distress early in HIV care was a predictor of future non-adherence to antiretroviral treatment, emphasising the need for early detection and intervention for psychological distress in people living with HIV [73]. The study advocated for integrating MH interventions with adherence strategies to improve HIV treatment outcomes.

Patients refraining from taking responsibility for their own health: The Free State managers observed that TB patients with MH disorders often refuse both MH and TB treatments, underscoring a significant barrier to integrating MH services into TB care. This issue reflects broader concerns about the expectation that individuals should be solely accountable for their adverse health outcomes. Levy (2019) [74] challenged this notion by arguing that it forms an inadequate foundation for policy. The capacity to make responsible health choices is often inequitably distributed, influenced by social and economic disparities. Therefore, it is flawed to place sole responsibility on individuals for making poor health choices. Instead, accountability should rest with those who influence the distribution of resources and shape broader circumstances, as they indirectly contribute to adverse health outcomes and also possess the means to address these underlying inequities.

Moreover, the low level of MH literacy among the South African population [75] exacerbates this issue. Without an adequate understanding of MH, individuals are less likely to recognise symptoms, seek help, or adhere to treatment plans. This lack of literacy further undermines the expectation that patients should take full responsibility for their health outcomes. Improving MH literacy is essential to empower individuals to make informed health decisions and to support the integration of MH services into TB and HIV care effectively [76]. Wiedermann et al. (2023) [77] presented well-founded recommendations to bolster MH support within educational systems, especially during crises like the COVID-19 pandemic. These authors champion a proactive and comprehensive approach to MH, suggesting the integration of MH education into the core curriculum to help students develop coping mechanisms and emotional intelligence. Additionally, they emphasised the necessity of training educators to recognise and address MH issues, and advocate for interdisciplinary collaboration among various stakeholders.

#### 4.7.2. People Facilitators

Increasing family involvement: The managers in the current study identified increasing family involvement as a crucial facilitator for the integration of MH services, recognising the significant role families play in providing emotional and practical support. This involvement creates a nurturing environment that enhances treatment adherence and overall well-being. Evidence from an Ethiopian study supports this view, showing that TB/HIV co-infected patients had a lower quality of life across all domains compared to patients with HIV but without active TB. The study found that strong family support was significantly associated with improvements in most quality-of-life domains [78].

Partnering with traditional and religious leaders: Recent findings from diverse settings—including Ghana, India, the occupied Palestinian territories, and South Africa—highlight that essential MH resources extend beyond formal healthcare systems to include religious groups, and faith-based organisations [68]. The Free State programme managers believed that partnering with traditional and religious leaders is crucial for the effective integration of MH services into TB and HIV programmes. In Sub-Saharan Africa, there is a widespread belief that traditional healers have special skills to diagnose and treat both physical and emotional ailments stemming from social misconduct, spirits, spells, and sorcery—domains where allopathic doctors are often seen as inadequate [79]. In South Africa, traditional healers are also often seen to be more accessible than biomedical providers and play a vital role in addressing emotional and spiritual well-being.

Recognising the paucity of allopathic MH care providers in South Africa’s public sector (for example, only 0.4 psychiatrists and 0.3 psychologists per 100,000 people at the time), Audet et al. (2017) [79] commended innovative solutions, such as task sharing models being proposed in the government’s MHPF&SP 2013–2020 [20]. These authors however emphasised that despite these innovative policies, challenges persisted in implementing allopathic–traditional MH care integration. Van Rooyen et al.’s (2015) [80] (p. 4) research highlighted that the negative attitudes of allopathic health practitioners towards traditional healers emanated from “unscientific methods” used by the traditional health practitioners in treating MH patients, the “interference” of traditional healers with the efficacy of hospital MH treatment, and “delays” by traditional healers in referring MH patients to hospital.

Campbell-Hall et al. (2010) [81] emphasised the importance of creating collaborative models that integrate traditional and Western MH care systems in South Africa. These authors explored the perceptions of service users and providers in a rural sub-district of KwaZulu–Natal province and proposed the following steps to foster a functional relationship between these two systems. Firstly, adopting a collaborative approach that respects the domains of both traditional and Western practices, finding mutually reinforcing intersections. Secondly, Western practitioners should adopt a “meaning-centred” approach, which involves understanding the user’s perspective and negotiating treatment plans that may include traditional methods. Thirdly, addressing the need for alternative arrangements to bring together different MH service providers, suggesting the establishment of a multisectoral MH advisory group to facilitate integration at district and sub-district levels. Such groups should include representatives from traditional practitioners to ensure participatory processes.

### 4.8. Limitations

While qualitative studies, including FGDs and SSIs with public health programme managers, offer valuable insights, they are not without limitations. A notable limitation is the restricted generalisability of findings to other contexts or provinces. Qualitative research, relying on the subjective perspectives and experiences of participants, is susceptible to bias from both participants and researchers. The quality of data can vary based on factors such as participants’ willingness to share, the skill of the facilitator or interviewer, and group dynamics. In FGDs, dominant voices may overshadow others, leading to a partial or skewed understanding of the issues. Furthermore, the interpretive nature of qualitative research poses challenges for reproducibility and validation, as various researchers may differently interpret findings.

According to de Savigny and Adam [41], a health system cannot function effectively if its components are considered in isolation. Instead, it is the complex relationships and dynamic interactions among these components—how each one influences and is influenced by the others—that create an integrated and functional system. To fully understand a health system, one must examine both the configuration and interplay of its components, and how these interactions collectively enable the system to achieve its intended goals. The current exploratory study falls short of this comprehensive perspective as it primarily identifies key perceived barriers and facilitators to integrating MH services into TB and HIV services. However, it lays the groundwork for future research that could more deeply investigate the cross-cutting effects of these barriers and explore strategies to address them. Future qualitative research could, for example, examine how managerial perspectives influence the implementation of integrated MH–TB–HIV interventions, such as by promoting awareness and encouraging the routine use of integrated TB–HIV–MH screening tools.

## 5. Conclusions

This study highlights significant gaps in the integration of MH services with TB and HIV care in South Africa’s Free State province. Programme managers identified substantial structural barriers, such as inadequate MH management frameworks, and social obstacles, including stigma and insufficient family support for treatment adherence. Despite these challenges, the study emphasises the crucial role of robust management structures and integrated screening processes as key facilitators for successful integration.

The Free State’s underperformance in MH care, evidenced by a notably high rate of MH separation, underscores the urgent need for a comprehensive, system-wide strategy. Such a strategy must address both structural and social barriers that hinder MH integration. In the province’s public health sector, where the integration of MH services with TB and HIV care remains inadequate, a fragmented approach is unlikely to yield significant improvements. The province’s suboptimal TB outcomes and insufficient HIV testing coverage indicate deeper systemic interdependencies that require a cohesive, coordinated response.

A whole-system approach is imperative, conceptualising the healthcare system as an interconnected network where changes in one area inevitably affect others. By addressing all six building blocks of the health system and emphasising the central role of “people”—both as healthcare providers and recipients—this approach offers a more comprehensive and effective strategy for tackling the root causes of integration challenges.

## Figures and Tables

**Table 1 ijerph-21-01528-t001:** Subthemes highlighting managers’ perceptions of barriers to and facilitators of integration of MH into TB and HIV care.

Building Block	Barrier	Facilitator
Service delivery	Lack of MH support for diseases other than HIV Limited role of NGOs and CBOs Communication gaps between initiating and receiving facilities during down referral	Decentralise and deinstitutionalise MH care Establish a proper MH management structure Establish support groups for TB-HIV-MH patients
Workforce	Deficits in staff MH training Staff not prioritising MH Lack of district MH specialist teams	Improve staff MH training and retraining Re-establish district MH specialist teams
Health information	Unawareness/non-use of TB-HIV-MH screening tool	Integrate screening for TB, HIV, and MH Formalise inclusion of MH data elements into TB-HIV-MH screening
Essential medicines	Lack of nutritional support for patients on treatment	Utilise CHWs to distribute medicine to patients’ homes
Financing	Insufficient resources and lack of dedicated funding for MH	Utilise NGOs and CBOs more effectively
Leadership/ governance	Limited MH management structure and lack of subdistrict MH managers	Adopt a multisectoral approach to improve training of MH care providers
People	Stigmatisation Lack of family treatment adherence support Patients refraining from taking responsibility for their own health	Increase family involvement Partner with traditional and religious leaders

Abbreviations: CBO, community-based organisation; CHW, community health worker; MH, mental health; NGO, non-governmental organisation; TB, tuberculosis.

## Data Availability

Data are not available as this would compromise respondents’ anonymity.

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
