# Peer review of "Exploring Managers’ Insights on Integrating Mental Health into Tuberculosis and HIV Care in the Free State Province, South Africa"

_ijerph, 2024, doi:10.3390/ijerph21111528_

Round 1

Reviewer 1 Report

Comments and Suggestions for Authors

Firstly, I would like to thank you very much for the opportunity to review the article -  Exploring managers' insights into the integration of mental health into tuberculosis and HIV care in the Free State Province, South Africa

In the introduction there are lack onf references…ex.  Page 1 line 32 - Health managers often face significant obstacles, such as poor information sharing, a lack of time and personnel to spur programs, and confusion over who is responsible for patient care.; Page 1 line 41 - Decentralization facilitates the adaptation of integration processes to suit local contexts.

In general, the article deals with a complex topic, but it is well-founded, explaining the evolution of policies and their orientation to respond to new challenges, but there are some issues that can be improved:

Page 4 line 166 the authors cite table 1 as a reference  but this is only found on page 6 in the methodology. I suggest you review this reference and whether it is really necessary at this point, as it is used to analyze the barriers and facilitators of integration...

Page 4 lines 177 until 203 and following pages, see if all the acronyms are explained, or another way to explain what is intended as the text is too confusing for an international reading....

The results are presented focusing on what people said about the items , but it would be important that in the end of the presentation of each item the authors introduced a brief analytical and reflective summary of these findings and explained them  (briefly). Focus  the most important ideas, that are revel by respondents. This part will have to be reconfigured...

I suggest using some parts of the discussions and placing only the most pertinent and evident in the discussion of the results.

The discussion must respond to the objectives of the article by discussing the main findings. This is very long and should be presented as a whole, with running text highlighting only the large dimensions...

The discussion is very long and needs to be shortened by showing the reader the main findings and the conclusions of the article must be clear. It is important to reveal the article's contributions to an international audience.

I wish you good work

Author Response

Response to Reviewer 1 Comments

1. Summary

2. Questions for General Evaluation

Reviewer’s Evaluation

Response and Revisions

Does the introduction provide sufficient background and include all relevant references?

Yes

Are all the cited references relevant to the research?

This question does not appear in the review information

Is the research design appropriate?

Yes

Are the methods adequately described?

Yes

Are the results clearly presented?

Must be improved

An effort has been made to present the results more clearly – please see our response to Comment 4.

Are the conclusions supported by the results?

Can be improved

The conclusion has been revised in accordance with revised results.

3. Point-by-point response to Comments and Suggestions for Authors

Comments 1: In the introduction there are lack of references…ex.  Page 1 line 32 - Health managers often face significant obstacles, such as poor information sharing, a lack of time and personnel to spur programs, and confusion over who is responsible for patient care.; Page 1 line 41 - Decentralization facilitates the adaptation of integration processes to suit local contexts.

Response 1: Thank you for pointing this out. We added references for the first sentence – page 2, paragraph 1, line 47. The second sentence has been removed.

Comments 2: In general, the article deals with a complex topic, but it is well-founded, explaining the evolution of policies and their orientation to respond to new challenges, but there are some issues that can be improved: Page 4 line 166 the authors cite table 1 as a reference  but this is only found on page 6 in the methodology. I suggest you review this reference and whether it is really necessary at this point, as it is used to analyze the barriers and facilitators of integration...

Response 2: We agree and have removed the reference to Table 1 – page number 4, paragraph 4, line 186.

Comments 3: Page 4 lines 177 until 203 and following pages, see if all the acronyms are explained, or another way to explain what is intended as the text is too confusing for an international reading....

Response 3: We agree and have done away with the following acronyms and abbreviations here and also throughout the manuscript: MHC, MHD, DSTB, MDRTB, XDRTB, ART, PLWH and NDoH.

Comments 4: The results are presented focusing on what people said about the items, but it would be important that in the end of the presentation of each item the authors introduced a brief analytical and reflective summary of these findings and explained them (briefly). Focus the most important ideas, that are revel by respondents. This part will have to be reconfigured...

Response 4: We agree and have accordingly included a brief analytical and reflective summary focusing on the most important ideas for each barrier and facilitator subset of the results:

3.1.1. Service delivery barriers – page 8, paragraph 1, lines 334-337.

3.1.2. Service delivery facilitators – page 8, paragraph 8, lines 363-365.

3.2.1. Workforce barriers – page 9, paragraph 8, lines 391-398.

3.2.2. Workforce facilitators page 9, paragraph 8, lines 411-416.

3.3.1. Health Information barriers – page 9, paragraph 11, lines 427-430.

3.3.2. Health information facilitators – page 10, paragraph 4, lines 444-450.

3.4.1. Essential medicines barriers – page 10, paragraph 6, lines 461-464.

3.4.2. Essential medicines facilitators – page 10, paragraph 10, lines 472-474.

3.5.1. Financing barriers – page 11, paragraph 1, lines 483-485.

3.5.2. Financing facilitators – page 11, paragraph 4, lines 496-500.

3.6.1. Leadership/governance barriers – page 11, paragraph 7, lines 509-513.

3.6.2. Leadership/governance facilitators – page 11, paragraph 10, lines 522-526.

3.7.1. People barriers – page 12, paragraph 6, lines 551-558.

3.7.2. People facilitators – page 12, paragraph 11, lines 575-579.

Comments 5: I suggest using some parts of the discussions and placing only the most pertinent and evident in the discussion of the results.

Response 5: We agree and have accordingly cut the discussion section focusing on the main research findings – page 12, paragraph 12, lines 580-863.

Comments 6: The discussion must respond to the objectives of the article by discussing the main findings. This is very long and should be presented as a whole, with running text highlighting only the large dimensions...

Response 6: We agree and have accordingly cut the discussion section focusing on the main research findings – page 12, paragraph 12, lines 580-863.

Comments 7: The discussion is very long and needs to be shortened by showing the reader the main findings and the conclusions of the article must be clear. It is important to reveal the article's contributions to an international audience.

Response 7: We agree and have accordingly cut the discussion section focusing on the main research findings – page 12, paragraph 12, lines 580-863. The conclusion has been revised accordingly – page 19, paragraph 2, lines 890-908.

4. Response to Comments on the Quality of English Language

Point 1: Not applicable.

5. Additional clarifications

Not applicable.

Reviewer 2 Report

Comments and Suggestions for Authors Minor revisions   Thank you for the opportunity to review this work. Integration of mental health into existing healthcare infrastructure and services is a key priority to reducing the treatment gap in low- and middle-income countries. As such, the manuscript is timely and enriches the understanding of how a 'whole-system' approach can be implemented in a situationally and culturally-specific way. The manuscript is competently written.     Section-by-section comments:   Abstract is written well overall, although there are some areas for improvement:
  • The authors should include a problem statement;
  • State when the study was conducted;
  • State sample size.
  Introduction: The introduction is informative. The policy and healthcare context in South Africa is overviewed.   Some areas requiring clarification are: L. 28-30: Define integrated care and provide appropriate references for the statements made. The importance of integrated care should be justified with reference to the mental health treatment gap in LMICs, together with relevant WHO guidance. L. 58: Please explain and support with data the statement '“growing and ill-characterized burden”. L. 90: Provide examples of 'contextually appropriate MH indicators in PHC facilities'. L. 99: Reference for the quotation is missing.   Clarify the distinctiveness and novelty of the current study. Have there been similar qualitative studies with healthcare stakeholders in SA?   Research questions are missing.     Methods: State the total sample size. How many participants did each FG have? Four seems like a very small number. Please explain this choice.   Recruitment: Clarify if individuals freely chose to participate and that they were not coerced. Explain how many of the interviews and FGs were in person and how many - remote. What software was used?   Provide demographic information about the participants - gender, professional roles, level of experience etc.   Provide a topic guide: What questions were posed to participants?     Results: The findings are rich and concisely and logically presented overall. Quotes have been used effectively. At times, however, the quotes are relied on to do most of the explaining, with the authors' analysis being rather thin. This is evident in '3.2.2. Workforce facilitators' and '3.6.2. Leadership/governance facilitators'. For those sub-sections, the authors are advised to elaborate on their own analysis, achieve a greater level of synthesis, and use quotes more strategically and concisely.   Clarification is required about the 'Health Information' theme. The content seems more suitable for the 'Service delivery' theme. Were there insights specific to health information/surveillance systems?   The sub-theme about traditional healers appears to be more related to the 'People' or 'Service Delivery' themes. Please reconsider its placement and review WHO's building blocks framework (https://iris.who.int/bitstream/handle/10665/258734/9789241564052-eng.pdf ).   'People' theme: 'Patients refraining to take responsibility for their own health': It is good to see this finding being critically discussed in the Discussion.   Discussion: Comprehensive although repetition can be minimised at times. For example l. 711-735 are not needed. The building blocks are already referenced in the Introduction and the Discussion is not the place to describe them in detail. The same applies to l. 817-834. The building blocks could be concisely described in a new table in the Introduction although this is a suggestion only.   L. 517-525: References are needed to support the claims made. L. 686-693: References are needed.   'Partnering with traditional and religious leaders': It would be useful to quote SA examples of collaborations between allopathic and traditional practitioners (e.g. Campbell-Hall et al., 2010; https://doi.org/10.1177/1363461510383459 ).   'Patients refraining to take responsibility for their own health': What about mental health literacy? This sub-section could be more compelling and specific. For example, what is meant by 'creating supportive environments'? Can you provide examples of relevant initiatives or interventions?

Author Response

Response to Reviewer 2 Comments

1. Summary

Thank you very much for taking the time to review this manuscript. Please find the detailed responses below and the corresponding revisions/corrections highlighted in the re-submitted file.

2. Questions for General Evaluation

Reviewer’s Evaluation

Response and Revisions

Does the introduction provide sufficient background and include all relevant references?

Yes

Are all the cited references relevant to the research?

This question does not appear in the review information

Is the research design appropriate?

Yes

Are the methods adequately described?

Yes

Are the results clearly presented?

Yes

Are the conclusions supported by the results?

Yes

3. Point-by-point response to Comments and Suggestions for Authors

Comments 1: Abstract is written well overall, although there are some areas for improvement:

  • The authors should include a problem statement;
  • State when the study was conducted;
  • State sample size

Response 1: Thank you for pointing this out. We agree with these comments and have made the following changes:

·        The following sentence that has been added to the abstract as a problem statement: “The integration of mental health (MH) services into tuberculosis (TB) and HIV care remains a significant challenge in South Africa’s Free State province. This study seeks to understand the perspectives of public health programme managers on the barriers to such integration and to identify potential strategies to overcome these challenges – page 1, paragraph 1, lines 5-8.

·        The data gathering period, between February and October 2021is now indicated – page 1, paragraph 1, lines 8-9.

·        The sample size, “a total of 15 managers” is now indicated in the abstract – page 1, paragraph 1, line 10. Further details are provided in section “2.3 Participant sampling and recruitment:” “A total of 15 managers responsible for the MH, PHC, TB, and HIV programmes participated in the study: four in the individual SSIs and 11 in two FGDs respectively involving five TB programme and six HIV programme participants” – page 16, paragraph 6, lines 294-300.

Comments 2: Introduction: The introduction is informative. The policy and healthcare context in South Africa is overviewed. Some areas requiring clarification are: L. 28-30: Define integrated care and provide appropriate references for the statements made. The importance of integrated care should be justified with reference to the mental health treatment gap in LMICs, together with relevant WHO guidance. L. 58: Please explain and support with data the statement “growing and ill-characterized burden”. L. 90: Provide examples of 'contextually appropriate MH indicators in PHC facilities'. L. 99: Reference for the quotation is missing. Clarify the distinctiveness and novelty of the current study. Have there been similar qualitative studies with healthcare stakeholders in SA?

Response 2: We agree and have, accordingly, revised the introduction as follows:

Integrated care has been defined with appropriate references – page 1, paragraph 3, lines 32-40.

The importance of integrated care has been justified with reference to the mental health treatment gap in LMICs, together with relevant WHO guidance – page 1, paragraph 3, lines 27-29.

The statement that South Africa has a growing burden of NCDs has been supported with data – page 2, paragraph 3, lines 69-72.

Examples of 'contextually appropriate MH indicators in PHC facilities' in South Africa’ have been provided – page 3, paragraph 1, lines 105-107.

The reference for the quotation in line 99 has been provided – page 3, paragraph 2, line 114.

An attempt to motivate the significance and uniqueness of the study is included – page 3, paragraph 6, lines 146-157. We have added that as far as could be established this is the first qualitative study focusing on programme managers’ views on integration of MH into PHC services in South Africa – page 4, paragraph 1, lines 155-157.

Comments 3: Research questions are missing.

Response 3: The research questions have been added – page 4, paragraph 1, lines 153-155.

Comments 4: Methods: State the total sample size. How many participants did each FG have? Four seems like a very small number. Please explain this choice. Recruitment: Clarify if individuals freely chose to participate and that they were not coerced. Explain how many of the interviews and FGs were in person and how many - remote. What software was used? Provide demographic information about the participants - gender, professional roles, level of experience etc. Provide a topic guide: What questions were posed to participants?

Response 4: We agree and have accordingly made the following changes in the methods section:

The total sample size and the number of participants in each FGD are now stated – page 6, paragraph 6, lines 294-300.

The voluntary and confidential nature of participation is now indicated – page 5, paragraph 5, lines 247-248.

The data collection section now explains that while both of the FGDs were conducted in person, two of the SSIs took place remotely – page 6, paragraph 3, lines 2769-271.

The qualitative analysis was conducted manually, without the aid of software tools.

Comments 5: Results: The findings are rich and concisely and logically presented overall. Quotes have been used effectively. At times, however, the quotes are relied on to do most of the explaining, with the authors' analysis being rather thin. This is evident in '3.2.2. Workforce facilitators' and '3.6.2. Leadership/governance facilitators'. For those sub-sections, the authors are advised to elaborate on their own analysis, achieve a greater level of synthesis, and use quotes more strategically and concisely. Clarification is required about the 'Health Information' theme. The content seems more suitable for the 'Service delivery' theme. Were there insights specific to health information/surveillance systems? The sub-theme about traditional healers appears to be more related to the 'People' or 'Service Delivery' themes. Please reconsider its placement and review WHO's building blocks framework (https://iris.who.int/bitstream/handle/10665/258734/9789241564052-eng.pdf). 'People' theme: 'Patients refraining to take responsibility for their own health': It is good to see this finding being critically discussed in the Discussion.

Response 5: Agree, we have, accordingly revised the results as follows:

We added summary interpretations after presenting the results for ‘3.2.2 Workforce Facilitators – page 9, paragraph 7, lines 410-415 and after the results for ‘3.6.2 Leadership/governance facilitators – page 11, paragraph 9, lines 521-525. Note that as per the comments of Reviewer 1, the same was done for all the other subsections of the results section. An attempt has been made to use the quotes more strategically and concisely throughout all the subsections of the results.

We would like to clarify the inclusion of the issue of “integrated screening for TB, HIV and MH and to formalise the inclusion of MH data elements into TB-HIV-MH screening” as a health information (rather than a service delivery) as follows. The section ‘3.3.2. Health information facilitators’ now includes the following summary/reflective sentence: “The main health-information-related facilitator for integrating MH into TB and HIV care perceived by the managers was the need for consistent combined screening for TB, HIV, and MH at PHC facilities using the available tool that many healthcare workers do not seem to be aware of. Additionally, formalising the inclusion of MH data elements in these screenings was seen to be crucial for consistency. Participants highlighted the need for awareness and formal processes to ensure effective integration and consistent care.” – page 10, paragraph 3, lines 443-449. We believe that this addition clarifies that the issues raised are indeed health information-related.

Having considered the WHO building blocks framework, we agree that the sub-theme about traditional healers appears is more related to the 'People' themes and have shifted it accordingly in Table 1 (page 7, paragraph 1, lines 303-306), the results (page 12, paragraph 8, lines 556-572) and discussion (page 17, paragraph 6, lines 828-862) sections.

Comments 6: Discussion: Comprehensive although repetition can be minimised at times.

For example l. 711-735 are not needed. The building blocks are already referenced in the Introduction and the Discussion is not the place to describe them in detail. The same applies to l. 817-834. The building blocks could be concisely described in a new table in the Introduction although this is a suggestion only. L. 517-525: References are needed to support the claims made. L. 686-693: References are needed. 'Partnering with traditional and religious leaders': It would be useful to quote SA examples of collaborations between allopathic and traditional practitioners (e.g. Campbell-Hall et al., 2010; https://doi.org/10.1177/1363461510383459 ). 'Patients refraining to take responsibility for their own health': What about mental health literacy? This sub-section could be more compelling and specific. For example, what is meant by 'creating supportive environments'? Can you provide examples of relevant initiatives or interventions?

Response 6: We agree and have revised the discussion as follows:

Lines 711-735, 817-834, 517-525 and all the other general descriptions of the building blocks in the Discussion section have been removed.

A reference has been added for the statement in line 686 in the reviewed manuscript (now page 14, par 6, line 684). The rest of the text for which no references could be found has been removed.

The section on ‘Partnering with traditional and religious leaders’ has been expanded with addition of reflections on collaborations between allopathic and western MH care providers, based on Campbell-Hall et al. (2010) as well as two later studies: Van Rooyen et al. (2015) and Audet et al. (2017) page 18, paragraph 1, lines 833-836 and page 18, paragraph 2, lines 839-862.

The section on 'Patients refraining to take responsibility for their own health' has been revised to be more compelling and specific – page 17, paragraph 4, lines 806-818.

The text referring to ‘creating supportive environments’ has been removed.

4. Response to Comments on the Quality of English Language

Point 1: Not applicable.

5. Additional clarifications

Not applicable.